# Respiratory depression and analgesia by opioid drugs in freely behaving larval zebrafish

Shenhab Zaig[1], Carolina da Silveira Scarpellini[2], Gaspard Montandon[1]*

[1]Keenan Research Centre for Biomedical Sciences. St. Michael's Hospital Unity Health Toronto, Toronto, Canada; [2]Department of Medicine, Faculty of Medicine, University of Toronto, Toronto, Canada

**Abstract** An opioid epidemic is spreading in North America with millions of opioid overdoses annually. Opioid drugs, like fentanyl, target the mu opioid receptor system and induce potentially lethal respiratory depression. The challenge in opioid research is to find a safe pain therapy with analgesic properties but no respiratory depression. Current discoveries are limited by lack of amenable animal models to screen candidate drugs. Zebrafish (*Danio rerio*) is an emerging animal model with high reproduction and fast development, which shares remarkable similarity in their physiology and genome to mammals. However, it is unknown whether zebrafish possesses similar opioid system, respiratory and analgesic responses to opioids than mammals. In freely-behaving larval zebrafish, fentanyl depresses the rate of respiratory mandible movements and induces analgesia, effects reversed by µ-opioid receptor antagonists. Zebrafish presents evolutionary conserved mechanisms of action of opioid drugs, also found in mammals, and constitute amenable models for phenotype-based drug discovery.

## Introduction

The current North American opioid epidemic is staggering, with millions of emergency visits and over 45,000 deaths annually (*Florence et al., 2016*). Synthetic opioids like fentanyl and oxycodone, or natural opioids like heroin and morphine are highly addictive (*Wilkerson et al., 2016*) and can lead to respiratory depression (*Dahan et al., 2010*; *Montandon et al., 2016a*; *Nagappa et al., 2017*), that can be lethal with overdose (*Gomes and Juurlink, 2016*; *Jones et al., 2013*). Opioids are critical pain therapies and there are currently no other medications that offer effective relief of pain (*IASP, 2018*). However, the side-effects of opioids limit their effective use and may lead to suboptimal pain treatments.

Opioid medications act on µ-, δ-, and κ- opioid receptors (*Gutstein, 2001*), which are expressed in discrete brain circuits. Although a wide range of opioids targeting these opioid receptors exist, the most potent opioids, such as fentanyl and oxycodone, are largely selective for the µ-opioid receptors (MORs) (*Sora et al., 1997*). Because MORs are expressed in neural circuits involved in breathing and nociception, opioids present effects beyond their intended therapeutic analgesic properties. MOR drugs induce side-effects such as addiction (*Le Merrer et al., 2009*), sedation (*Montandon et al., 2016a*), and hypoventilation (*Dahan et al., 2010*; *Montandon and Slutsky, 2019*; *Nagappa et al., 2017*). Hypoventilation is characterized by reduced respiratory rate (*Macintyre et al., 2011*), reduced airflow (*Ferguson and Drummond, 2006*), prolonged apneas (*Mogri et al., 2008*; *Nagappa et al., 2017*), and severe hypoxemia (*Brown et al., 2006*). Complete respiratory arrest and bradycardia are the main causes of death by opioid overdose (*Pattinson, 2008*). Importantly, respiratory depression and cessation of breathing with opioid overdose are due to the action of opioid drugs on brainstem respiratory network (*Montandon and Slutsky,*

*For correspondence:
gaspard.montandon@utoronto.ca

Competing interests: The authors declare that no competing interests exist.

**eLife digest** When it comes to treating severe pain, a doctor's arsenal is somewhat limited: synthetic or natural opioids such as morphine, fentanyl or oxycodone are often one of the only options available to relieve patients. Yet these compounds can make breathing slower and shallower, quickly depriving the body of oxygen and causing death. This lethal side-effect is particularly devastating as opioids misuse has reached dangerously high levels in the United States, creating an 'opioid epidemic' which has claimed the lives of over 80,000 Americans in 2020. It is therefore crucial to find safer drugs that do not have this effect on breathing, but this research has been slowed down by the lack of animal models in which to study the effect of new compounds.

Zebrafish are small freshwater fish that reproduce and develop fast, yet they are also remarkably genetically similar to mammals and feature a complex nervous system. However, it is not known whether the effect of opioids on zebrafish is comparable to mammals, and therefore whether these animals can be used to test new drugs for pain relief.

To investigate this question, Zaig et al. exposed zebrafish larvae to fentanyl, showing that the fish then exhibited slower lower jaw movements – a sign of decreased breathing. The fish also could also tolerate a painful stimulus for longer, suggesting that this opioid does reduce pain in the animals. Together, these results point towards zebrafish and mammals sharing similar opioid responses, demonstrating that the fish could be used to test potential pain medications. The methods Zaig et al. have developed to establish these results could be harnessed to quickly assess large numbers of drug compounds, as well as decipher how pain emerges and can be stopped.

*2019*). The only available treatment to reverse respiratory depression during an opioid overdose is the MOR antagonist naloxone which directly blocks the binding of opioid ligands to MORs. However, naloxone also blocks the analgesic properties of opioid drugs, so it cannot be used as an adjunct treatment. The discovery of novel opioid drugs or combinations of drugs with potent analgesic properties without the side-effects of respiratory depression has been hindered due to the lack of animal models allowing behavioral assessments of pain and breathing, while allowing for large-scale drug discovery.

Identification of new molecular targets and drugs is critical to the development of safe opioid therapies. Due to the complexity of the machinery regulating MOR inhibition, wide scale screening of drugs acting on the known mechanisms regulating MOR inhibition in rodent models is not feasible. It is therefore critical to identify amenable model systems where large-scale screening can be developed, while preserving the complexity of vertebrate central nervous system and behaviours. The challenge is to quantify complex phenotypes like respiratory neural activity and nociception on a large scale, which is normally limited by the complexity of the animal models. Rodents are not optimal because their large size, generation time, and drug availability limit screen scale. An alternative is the larval zebrafish, which shares anatomy, physiology, and large parts of the genome with humans. The small size of larval zebrafish, ease of production, and their external fertilization which allows easy gene knockdown, makes the zebrafish an attractive model organism to screen genes and drugs. Here, we propose to use the larval zebrafish as simple, yet powerful, animal model, so that the molecular mechanisms of MOR respiratory and nociceptive inhibition can be identified. Importantly, the zebrafish larvae can also be used for large-scale drug screening to identify new drug candidates.

The larval zebrafish has emerged as an ideal model system for drug and gene discovery since it combines the biological complexity of in vivo models with a nervous system homolog to humans including a respiratory neural network. Although fish use a different strategy to absorb oxygen and eliminate carbon dioxide than mammals, they rhythmically produce mandibular movements to move water through their gills. In lampreys, the respiratory neural network includes the paratrigeminal respiratory group (pTRG) and the vagal nucleus and this network generates rhythmic mandible movements (*Bongianni et al., 2016*). Because of their close evolutionary origins (*Missaghi et al., 2016*), the respiratory network shares similarities with the mammalian respiratory network (*Cinelli et al., 2013*) which generates breathing and regulates respiratory depression by opioids (*Montandon et al., 2011*). Interestingly, the pTRG presents similar functional properties than the

mammalian respiratory network such as sensitivity to substance P (*Mutolo et al., 2010*) and to opioid ligands (*Mutolo et al., 2007*). Here, we propose that respiratory mandible movements can be used as an index of respiratory network activity, similar to respiratory activity of the trigeminal muscle in mammals (*Jacquin et al., 1999*). The zebrafish also possesses a nociceptive system encompassing spinal cord, brainstem and sub-cortical circuits (*Taylor et al., 2017*), with homology to mammals. Although pain circuit activity and its response to opioids cannot be directly assessed, the swimming escape response to nociceptive stimuli can be easily assessed in larval zebrafish. Nociceptive stimuli such as formalin (*Magalhães et al., 2017*) or (allyl)-isothiocyanate (AITC, also known as 'mustard oil') can be administered to larval zebrafish. AITC acts on transient receptor potential ankyrin 1 channels and transient receptor potential cation channel vanilloid receptor 1 (*Oda et al., 2016*) which are involved in nociception to chemical compounds (*Bamps et al., 2021*). Here, we propose to determine whether respiratory depression and analgesia can be measured in larval zebrafish.

Most opioid analgesics and drugs of abuse elicit their effects by binding to MORs (*Gutstein, 2001*). In zebrafish, the MOR has high homology to the mammalian MOR and shares 74% of its amino acid sequence with its mammalian counterpart (*Sanchez-Simon and Rodriguez, 2008*). The MOR also possesses similar binding properties to the mammalian MOR for morphine and DAMGO (*Marron Fdez de Velasco et al., 2009*; *Mutolo et al., 2007*). Because of the homologies of the respiratory, pain, and opioid systems in zebrafish and mammals, we propose that the larval zebrafish may mimic respiratory depression and analgesia by opioids and other phenotypes associated with opioid drugs in mammals. The objectives of this study are to demonstrate that larval zebrafish have evolutionary conserved opioid properties mimicking mammals and that they can be used to investigate opioid-induced respiratory depression and analgesia. We aim to demonstrate that larval zebrafish replicates the opioid pharmacology observed in mammals and humans and that current pharmacotherapies to block respiratory depression can be tested in larval zebrafish.

## Results

Zebrafish use gills to promote gas exchange with water. Flow of water through the gills is generated by a complex respiratory network in the brainstem which presents similar anatomy and properties to the mammalian respiratory network (*Figure 1A*). The respiratory network generates rhythmic movements and rhythmically activate muscles which promotes water flow through the gills. One of these muscles, the mandible muscle controls opening and closing of the mandible and is activated by the mandible nerve which originate from the trigeminal nerve. Mandible movements can be easily recorded using a high-definition camera positioned on top of the swimming zebrafish (*Figure 1B,C*), and is a robust index of respiratory network activity. Mandible movements can be quantified by counting the changes in pixel intensity around the mandible area (*Figure 1B*) and can be displayed over time (*Figure 1D*). The rate of mandible movements, identified as the number of mandible movement peaks (red circles, *Figure 1D*), indicates respiratory network rhythm, and is considered here as an index of respiratory rate. Using the convenience of multi-well plates, we administered combinations of drugs to embryo medium as described in *Figure 1E*. Combinations of drugs were administered to separate groups of fish. Each animal only received one combination of drugs and did not receive consecutive combinations of drugs because it would not be feasible to remove drugs from the water.

To determine whether zebrafish can be used as a model of opioid-induced respiratory rate depression, we administered embryo water (control) or fentanyl - a commonly used opioid analgesic - to larval zebrafish while quantifying the rate of mandible movements (*Figure 2A*). Larval zebrafish under control conditions (embryo water) presented rates of mandible movements ranging from 15 to 115 movements per minute. Larvae were exposed to a concentration of fentanyl of 1 µM and respiratory rate was recorded over a 30 min time-period (*Figure 2B*). In the control group, respiratory rate initially increased compared to baseline but did not significantly change during the 30 min following baseline (*Figure 2B*). In the fentanyl group, respiratory rate was strongly reduced 4 min after fentanyl administration. Since the strongest respiratory rate depression was observed 5 min after exposure, we compared respiratory rates (measured over a one-minute time-period between minutes 5 and 6) of control and fentanyl groups (*Figure 2C*). Fentanyl (0.2, 0.4 µM) did not significantly decrease rate of mandible movements compared to control (p=0.473 and p=0.797,

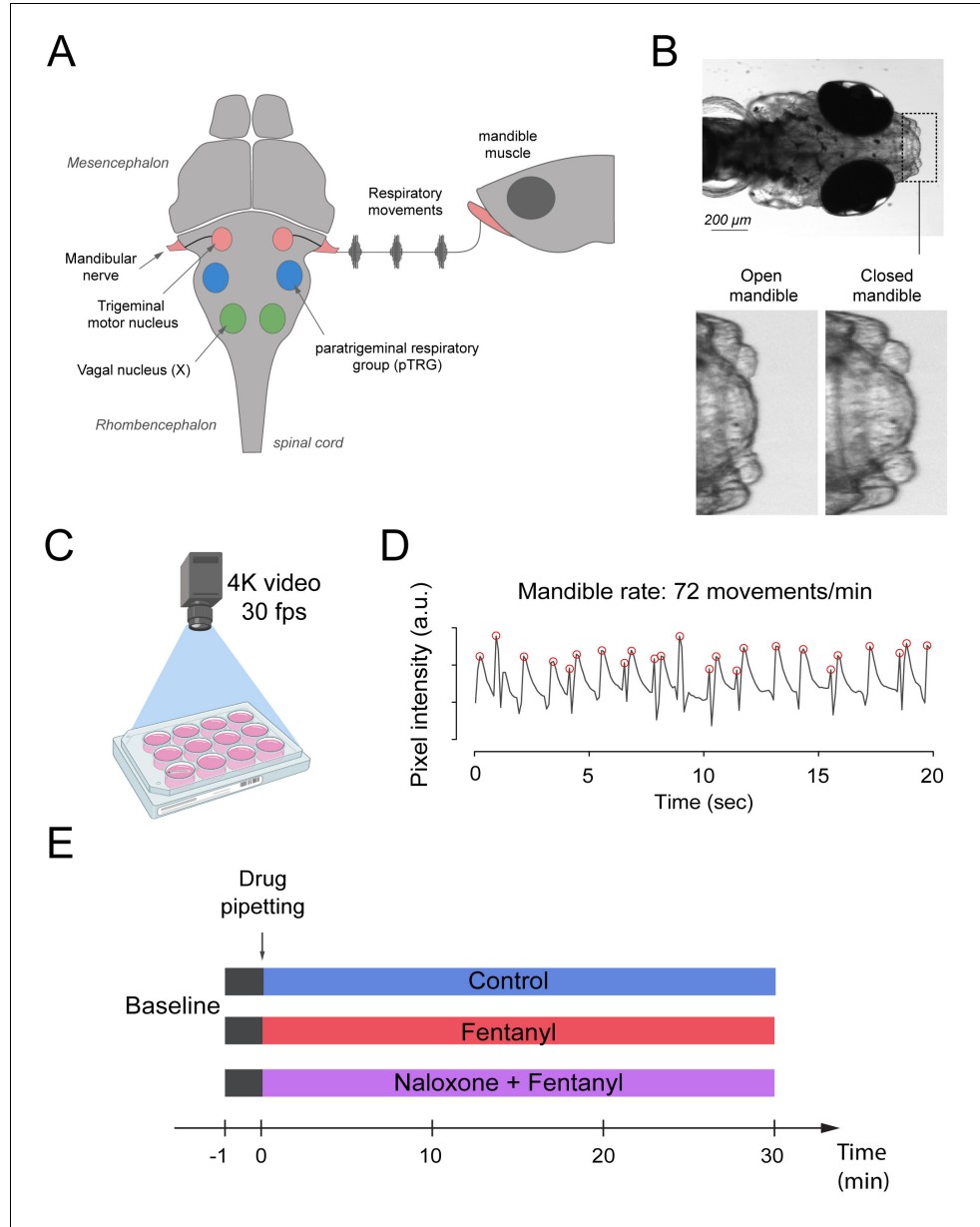

**Figure 1.** Respiratory network activity quantified using mandible movements in freely-moving larval zebrafish. (**A**) The brainstem respiratory network of the larval zebrafish produces respiratory mandible movements. (**B**) Respiratory mandible movements are used as an index of respiratory network activity. (**C**) Mandible movements were quantified by looking at the pixel changes in the region of the mandible using a 4K high-definition camera. (**D**) Respiratory rate was quantified in 12–14 day post-fertilization larvae positioned in 12-well plates. Red circles indicate peaks of mandible movements. (**E**) To quantify respiratory rate depression induced by opioid drugs, fish were exposed to a drug treatment containing embryo medium only (control), fentanyl, or a combination of naloxone and fentanyl.

*Figure 2C*). Fentanyl (1 and 3 µM) significantly decreased rate compared to control groups (p=0.006 and p=0.038, *Figure 2C*). When data were normalized according to baseline rate, variability within groups was significantly reduced (*Figure 2D*). Fentanyl significantly decreased normalized rate at 1 µM (p<0.001) and 3 µM (p<0.001) compared to the control group (*Figure 2E*), but not at 0.2 µM (p=0.275) and 0.4 µM (p=0.200). In summary, fentanyl-induced respiratory rate depression was observed at 1 and 3 µM of fentanyl tested. We determined that such normalization best represented

respiratory rate depression by fentanyl and we only presented normalized data for the subsequent figures with respiratory depression.

To determine the pharmacology of opioid receptors in zebrafish, we compared groups of fish exposed to fentanyl with fish exposed to fentanyl and the MOR antagonists naloxone and CTAP (*Figure 3*). Exposure to fentanyl reduced the rate of mandible movements (*Figure 3A*). The MOR antagonist naloxone at 10 µM did not significantly reverse respiratory rate depression induced by fentanyl (p=0.178, *Figure 3B*). However, naloxone (20 µM) significantly reversed respiratory rate depression by fentanyl (p<0.001, *Figure 3B*). The selective MOR antagonist CTAP (4 µM) did not significantly reverse respiratory rate depression by fentanyl (p=0.077, *Figure 3B*), although a trend toward significance was observed. To determine whether naloxone or CTAP may increase respiratory rate when administered alone, we compared groups of fish exposed to CTAP (4 µM) or naloxone (5 and 20 µM) to control fish (*Figure 3C*). Neither CTAP nor naloxone showed effects on respiratory rate when administered alone (one-way ANOVA, p=0.122). In summary, fentanyl induced a significant respiratory rate depression which was blocked by the MOR antagonist naloxone but not CTAP.

In the previous experiments, naloxone was administered concomitantly with fentanyl and prevented respiratory depression (*Figure 3A,B*). We then tested whether naloxone can completely reverse respiratory depression by fentanyl (1 µM) when administered after respiratory depression (*Figure 3D*). As expected, fentanyl significantly depressed respiratory rate (p=0.009), an effect not reversed by naloxone at 9 µM (p=0.111). Although a sequential approach may be of interest due to its repeated measure design, it presented two challenges. First, drugs were pipetted twice in the multi-well plate and affected the fish's behavior, which may explain the large variability observed with naloxone. Second, due to the low volume of fish water in wells and volumes of stock solution administered, it was challenging to increase naloxone concentration in these experiments without diluting the concentration of fentanyl. We therefore concluded that sequential administration of drugs was not the most consistent and compared drug combinations using separate groups of fish.

Because zebrafish strains may present genetic variability and different sensitivities to opioid drugs (*Marron Fdez de Velasco et al., 2009*; *Sanchez-Simon and Rodriguez, 2008*), we compared respiratory depression by fentanyl in various zebrafish lines (*Figure 3E*). Fentanyl (1 µM) significantly reduced respiratory rate in AB fish as previously determined in previous experiments (p<0.001), but not in Tübingen (TU) (p<0.001) or AB x TU zebrafish crosses (p<0.001). To conclude, only the AB zebrafish strain presented sensitivity to fentanyl and was used for subsequent experiments. In summary, respiratory rate was depressed by fentanyl through activation of opioid receptors, an effect reversed by naloxone.

Morphine is a widely-used opioid drug that induces less severe respiratory depression than the highly potent opioids such as fentanyl (*Gutstein, 2001*). We tested whether morphine induces respiratory rate depression in zebrafish. There was no dose-dependent decrease in respiratory rate with increasing morphine dosages (1, 10, 20, 50 and 200 µM, p=0.088, *Figure 3F*). However, multiple comparison tests showed that morphine at 1 µM significantly depressed respiratory rate (p=0.021) when compared to controls, but not at the other concentrations tested (p>0.488). These results suggest that morphine depressed respiratory rate at low concentration, but that at higher concentrations, morphine may induce an excitatory effect.

To determine whether breathing in zebrafish larvae can be stimulated by specific excitatory drugs as observed in mammals, we administered two respiratory stimulants previously shown to stimulate breathing when depressed by opioids. We first administered the 5-HT$_{4A}$ serotoninergic ligand BIMU8 (*Manzke et al., 2003*) with or without fentanyl (*Figure 4A*). BIMU8 (10 µM) combined with fentanyl did not show reversal of respiratory rate when compared to the fentanyl group (p=0.193). Fish exposed to BIMU8 alone showed significant higher respiratory rate compared to fentanyl alone (p=0.008) but was not different from the control group (p=0.294). BIMU8/fentanyl showed significantly lower respiratory rates compared to BIMU8 alone and control (p=0.018 and p=0.007). In summary, the addition of BIMU8 to fentanyl did not significantly reversed respiratory depression by fentanyl.

The AMPA positive allosteric modulator CX614 significantly reversed respiratory rate depression by fentanyl (p<0.001, *Figure 4B*). CX614/fentanyl showed higher respiratory rate than fentanyl alone, as well as higher respiratory rate than control (p=0.021), suggesting that CX614 at this dosage reversed respiratory depression by fentanyl. CX614 alone did not present higher respiratory

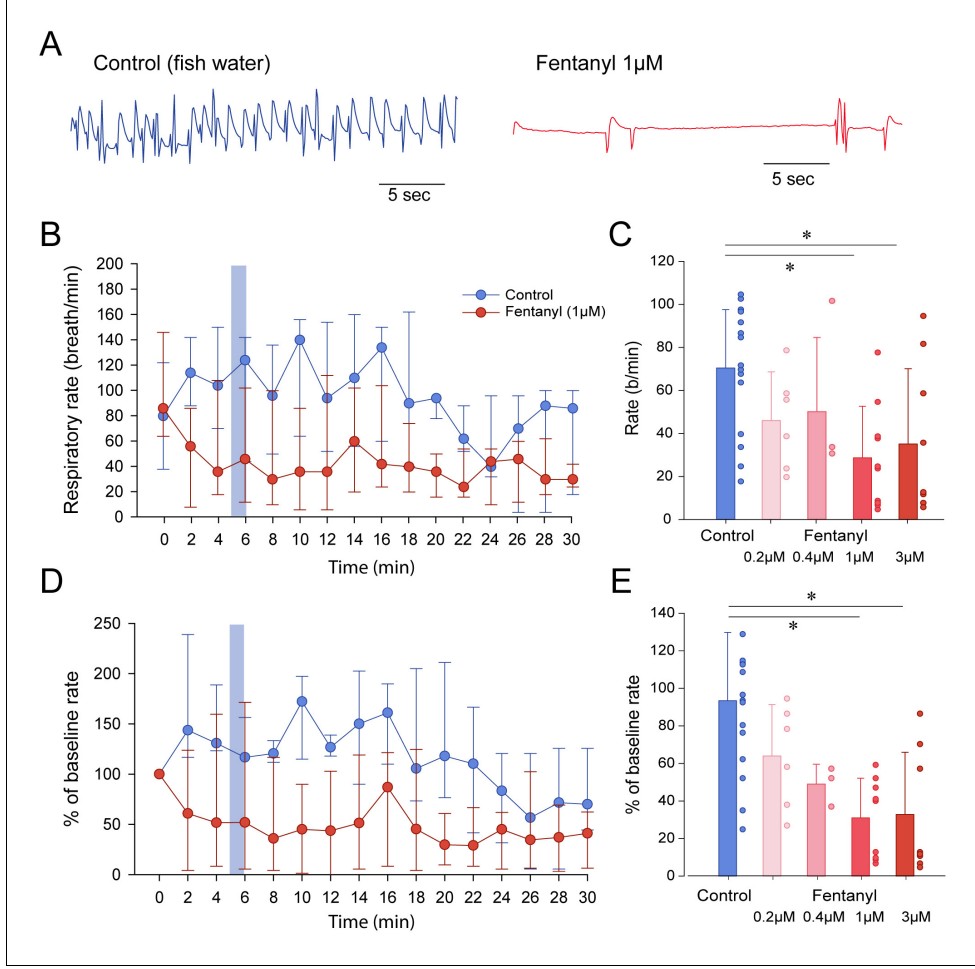

**Figure 2.** Respiratory rate depression by the opioid analgesic fentanyl in larval zebrafish. (**A**) Representative mandible movements in larvae exposed to embryo medium or a solution of fentanyl (1 μM). The rate of mandible movements was significantly decreased by fentanyl. (**B**) Larvae were exposed to fentanyl over a period of 30 min. Significant respiratory rate depression was observed within 4 min following fentanyl application. Respiratory rate data did not follow a normal distribution and data are presented as medians with bars representing interquartile range. (**C**) Increasing the concentration of fentanyl induced dose-dependent decreases in respiratory rate. Fentanyl produced significant decreases in respiratory rate at 1 μM (n = 9) and 3 μM (n = 8), but not at 0.2 μM (n = 6) and 0.4 μM (n = 3) compared to controls (n = 17). (**D**) Rates of mandible movements were not normally distributed due to the high variability in respiratory rates. To present more homogenous data, respiratory rates were normalized according to the baseline rate measured before drugs were applied to better represent how drug exposure changed rate. (**E**) Fentanyl produced significant decreases in respiratory rate at 1 μM and 3 μM, but not at 0.2 μM and 0.4 μM compared to controls. Shading indicates time periods used to calculate average data in panel **C** and **E**. In panels **B** and **C**, data were presented as medians ± 75th and 25th percentile. In panels **D** and **E**, data were presented as means ± standard deviations. Circles indicate individual data points for each zebrafish measured. * indicate significantly different medians compared to control with p<0.05. Source data can be found in *Figure 2— source data 1*.

The online version of this article includes the following source data for figure 2:

**Source data 1.** Respiratory rate depression by the opioid analgesic fentanyl.

rate when compared to control group (p=0.058) but showed a trend toward significance. Higher dosages of CX614 induced seizure and were not tested. To determine whether the respiratory depression observed with fentanyl was due to a direct effect of fentanyl on respiratory rate rather than an indirect effect on sedation (*Montandon and Horner, 2019*) or pain circuits (*Jiang et al., 2004*), we administered lidocaine, an analgesic not acting on MORs (*Lopez-Luna et al., 2017*). Lidocaine did not significantly change respiratory rate compared to control (p=0.267, *Figure 4C*). To

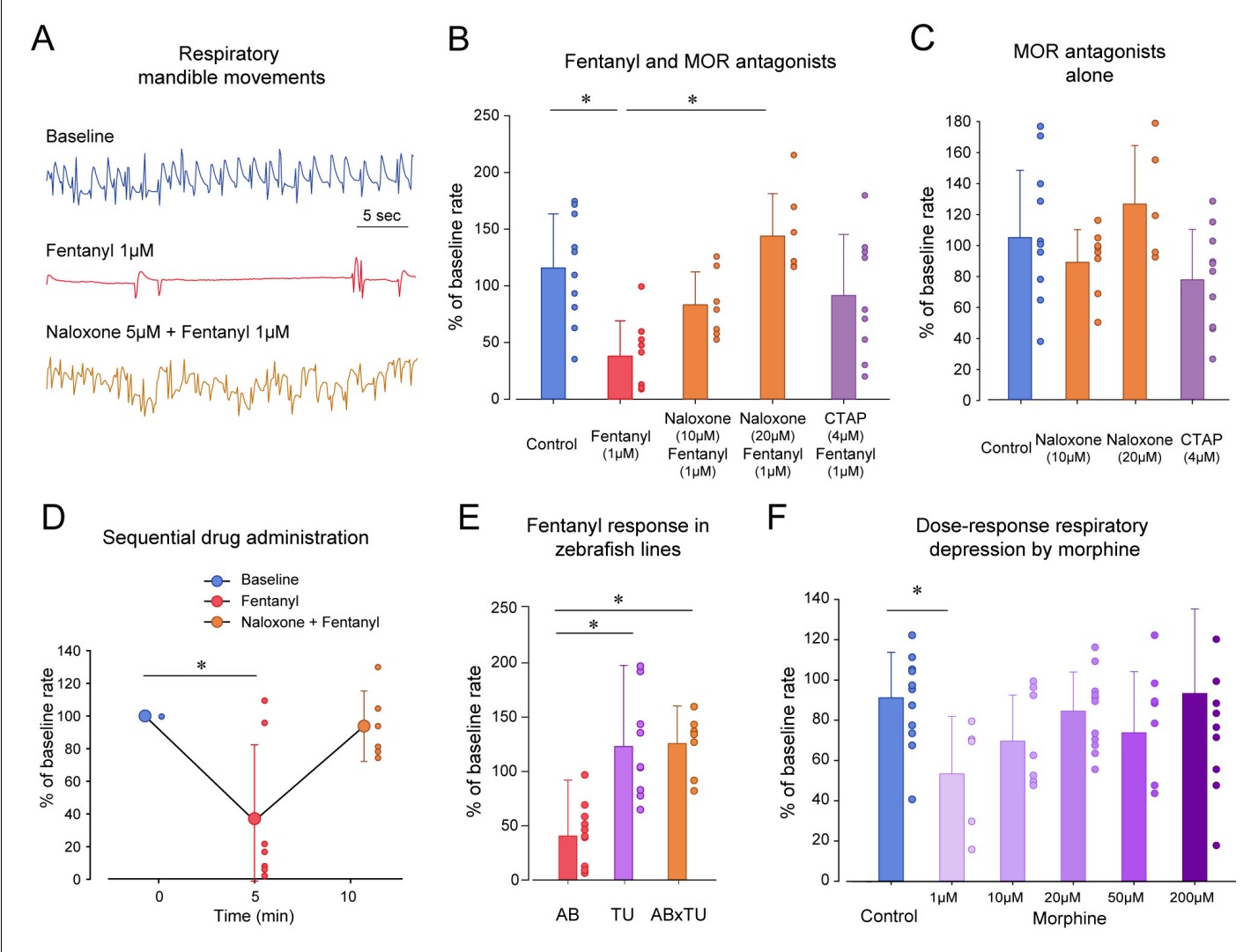

**Figure 3.** Opioid receptor pharmacology regulating respiratory depression in larval zebrafish. (**A**) Representative tracings of mandible movements in larval zebrafish showing fentanyl (1 μM) reducing respiratory rate and reversal by the MOR antagonist naloxone (20 μM). (**B**) Fentanyl (1 μM) significantly decreased rate of mandible movements (n = 9) compared to the control group (n = 10), an effect significantly reversed by the MOR antagonist naloxone (20 μM, n = 7) but not the highly selective MOR antagonist CTAP (4 μM, n = 9). (**C**) CTAP and naloxone administered alone did not affect respiratory rate. (**D**) Fentanyl depressed respiratory rate (n = 7), an effect reversed by subsequent addition of naloxone (9 μM). (**E**) Respiratory rate depression due to fentanyl was observed in the AB strain (n = 9), but not in TU (n = 9) or crosses between AB x TU (n = 7). (**F**) Morphine did not induce a dose-dependent decrease in respiratory rate compared to controls but showed a significant decrease at 1 μM. * indicates significantly different medians compared to control with p<0.05. Normalized data are presented as means ± standard deviations. Circles indicate individual data points for each zebrafish measured. AB, AB zebrafish line. TU, Tübingen zebrafish line. Source data can be found in *Figure 3—source data 1*.
The online version of this article includes the following source data for figure 3:

**Source data 1.** Opioid receptor pharmacology regulating respiratory depression.

dissolve drugs, we used DMSO as a solvent which may affect fish respiratory rate. DMSO administered at 0.0016% did not change respiratory rate (p=0.342).

To determine the analgesic properties of opioid drugs in larval zebrafish, we established a simple model of nociception combining exposure to formalin as a nociceptive stimulus and measurements of the subsequent swimming escape response. Using video recordings of larvae positioned in a multi-well plate, we tracked the individual movements of larvae exposed to embryo medium, formalin and a combination of formalin and fentanyl (*Figure 5*). We quantified two swimming behaviors (*Figure 5A*): swimming velocity (swimming distance in mm per second) and angular velocity (turn

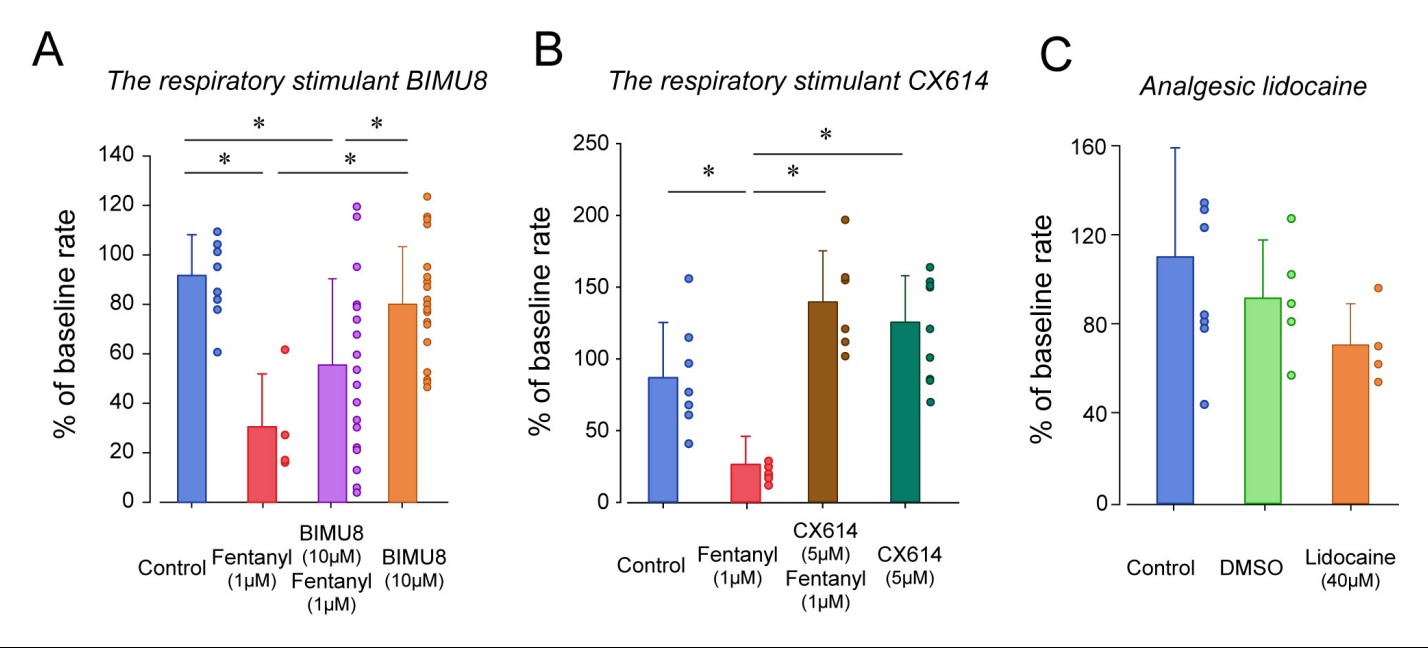

**Figure 4.** Respiratory depression and respiratory stimulants in larval zebrafish. (**A**) The respiratory stimulant BIMU8 (10 µM), a 5-HT$_{4A}$ serotonin receptor agonist, in combination with fentanyl (n = 20) was compared to fentanyl alone (n = 4) or BIMU8 alone (n = 21). BIMU8 was not sufficient to reverse respiratory depression by fentanyl. (**B**) The AMPA positive allosteric modulator CX614 (5 µM) and fentanyl (n = 6) were also compared to fentanyl alone (n = 7) or control (n = 7). CX614 + fentanyl group showed significantly higher respiratory rate than fentanyl, DMSO, and control. (**C**) The analgesic lidocaine (n = 4) was compared to DMSO (0.0016%) (n = 5) and control (n = 5) and showed lower respiratory rate than the control group. Normalized data are presented as means ± standard deviations. Circles indicate individual data points for each zebrafish measured. Black lines and * indicate groups significantly different with p<0.05. Source data can be found in *Figure 4—source data 1*.

The online version of this article includes the following source data for figure 4:

**Source data 1.** Respiratory depression and respiratory stimulants.

angle in degrees per second, *Figure 5—figure supplement 1*). In larvae exposed to formalin, swimming velocity was substantially increased by formalin (*Figure 5B*). Fentanyl combined with formalin showed considerable reduction in velocity compared to formalin alone (*Figure 5B*). Following exposure to formalin, swimming velocity significantly increased during the initial 3 min (*Figure 5C*, shaded blue), a response not observed in control larvae or formalin/fentanyl larvae (*Figure 5C*). We then compared the averaged swimming velocity during the first 3 min following drug exposure for the different groups of larvae. The formalin group presented significantly faster velocity compared to the control group (p<0.001, *Figure 5D*). Velocity in the formalin/fentanyl group was significantly slower than in the formalin group (p=0.037), suggesting that fentanyl reduced the escape response to formalin, which may indicate an analgesic effect. However, this effect may be due to the effects of fentanyl directly on swimming. Fentanyl administered alone did not decrease velocity when compared to control (p=0.221), showing that it did not affect swimming by itself. We then determined the effects of the µ-opioid receptor antagonists naloxone and CTAP (*Figure 5D*). In naloxone groups, velocities were not significantly higher than formalin/fentanyl (all p>0.122), suggesting that naloxone did not block the analgesic effects of fentanyl. In CTAP/formalin/fentanyl group, there was no difference compared to formalin/fentanyl or formalin alone (p=1.000 and p=1.000). Naloxone/formalin/fentanyl showed higher velocity than fentanyl or control groups (*Figure 5D*). These effects may be due to the impact of naloxone alone at 50 µM (*Figure 5E*). We then looked at the effect of formalin on angular velocity. The formalin group showed reduced angular velocity compared to control (p<0.001, *Figure 5—figure supplement 1*), an effect not reversed by fentanyl (p=0.151), which suggest angular velocity cannot be used to assess the escape response to formalin.

Although naloxone did not prevent or reverse to analgesic effects of fentanyl, it may be due to the large variability in swimming response induced by formalin. In fact, swimming velocity in formalin/fentanyl/naloxone ranged from 0.3 to 4 mm/sec, which is substantially larger than the variability

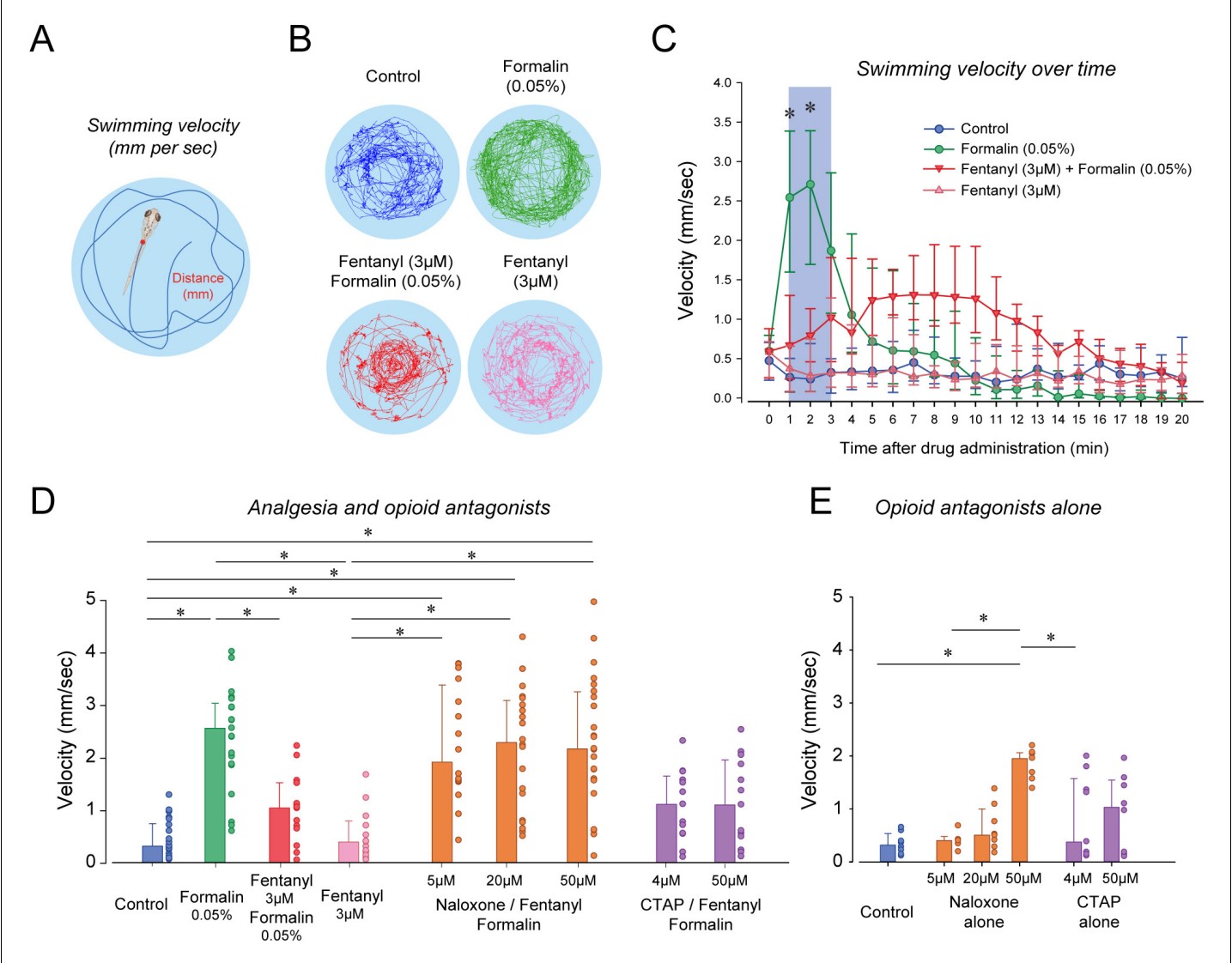

**Figure 5.** Nociception by formalin and analgesia by fentanyl in larval zebrafish. (**A**) As an index of nociception in larval zebrafish, we measured behavioral responses to the nociceptive stimulus formalin. We quantified swimming velocity in freely-moving larvae. Swimming velocity was quantified as the distance swam per second. (**B**) Control fish swam at a velocity ranging from 0.1 to 1.3 mm/sec, whereas swimming velocity was increased to 0.7–4.1 mm/sec with formalin. Addition of fentanyl reduced swimming velocity to 0.1–2.3 mm/s. (**C**) Following exposure to embryo medium only, formalin, or formalin/fentanyl, swimming velocity was strongly increased by formalin within 3 min and was only moderately increased by fentanyl and formalin compared to controls. (**D**) In separate groups of fish, comparison of averaged velocities for the first 3 min (shaded blue) after baseline (time = 0) showed that formalin (n = 21) significantly increased velocity compared to the control (n = 22), whereas fentanyl (n = 13) reduced the formalin response. Naloxone or CTAP did not block the effect of fentanyl at various concentrations. (**E**) Naloxone alone increased velocity at 50 µM, but not at 5 and 20 µM. CTAP alone did not increase velocity by itself. Data are presented as medians with error bars showing 25th and 75th percentile or interquartile range. Circles indicate individual data points for each zebrafish. * indicates groups significantly different with p<0.05. Source data can be found in *Figure 5—source data 1*.

The online version of this article includes the following source data and figure supplement(s) for figure 5:

**Source data 1.** Nociception by formalin and analgesia by fentanyl.

**Figure supplement 1.** Angular velocity in response to formalin and opioid drugs.

observed in control or fentanyl groups (*Figure 5D*). To induce more consistent nociceptive response, we administered (allyl)-isothiocyanate (AITC), a chemical acting AITC on transient receptor potential ankyrin 1 channels and transient receptor potential cation channel vanilloid receptor 1 (*Oda et al., 2016*). We used a similar experimental approach than we used with formalin by combining AITC,

fentanyl and/or naloxone. AITC (100 μM) induced a strong increase in velocity compared to controls, an effect reduced by fentanyl (*Figure 6A*). AITC produced an initial increase in swimming velocity during the first 3 min following drug exposure, which is consistent with the formalin response. Conversely, in fish with a combination of fentanyl (6 μM) and AITC (100 μM), no swimming response was observed compared to AITC alone (*Figure 6B*), showing that fentanyl reduced the nociceptive response to AITC. Compared to the concentration of fentanyl used in the formalin assay (3 μM), a higher concentration (6 μM) was necessary to eliminate the effects of AITC. This concentration of fentanyl (6 μM) did not decrease velocity when administered alone compared to controls (*Figure 6B*). Averaged velocities in AITC fish were significantly higher than in controls (p<0.001, *Figure 6C*), and this increase was significantly lower in fish exposed to fentanyl and AITC (p<0.001). Angular velocity was increased by AITC, but not reversed by fentanyl (*Figure 6—figure supplement 1*). Interestingly, groups with naloxone at 20 and 50 μM combined with AITC and formalin presented higher swimming velocities than groups exposed to AITC/fentanyl (p=0.003 and p<0.001, respectively), whereas CTAP did not (p=0.342). In summary, AITC produced a fast increase in swimming velocity, an effect significantly reduced by fentanyl, which was reversed by naloxone.

To determine whether larval zebrafish are sensitive to other types of analgesics, we exposed larvae to lidocaine, a non-opioid anesthetic (*Figure 7A*). As previously shown, formalin increased velocity compared to control (p<0.001, *Figure 7B*). In fish exposed to lidocaine and formalin, swimming velocity was significantly reduced compared to formalin group (p=0.002). Swimming velocity in lidocaine group was not different from control (p=0.693). Since we used a relatively high concentration of DMSO (1%) to dissolve lidocaine, we compared whether DMSO (1%) was different from control fish and we did not observed differences between the two groups (p=0.879, *Figure 7A*). Lidocaine did not affect angular velocity compared to formalin alone (*Figure 7—figure supplement 1*). In conclusion, two different types of analgesics (fentanyl and lidocaine) reduced the swimming response to formalin, therefore suggesting that these assays may properly assess analgesia.

The two stimulants BIMU8 and CX614 were administered to zebrafish to determine whether they can reverse respiratory depression by fentanyl. Only CX614 was able to significantly reverse respiratory depression. We therefore determine whether BIMU8 and CX614 were also reversing the analgesic properties of fentanyl. The combination of BIMU8 (20 μM)/formalin/fentanyl did not significantly change swimming velocity compared to the control (p=1.000) or formalin/fentanyl group (p=1.000, *Figure 7B*). However, CX614 (5 μM) /formalin/fentanyl group presented higher swimming velocities than the control group (p<0.001) and formalin/fentanyl (p=0.007), therefore showing that the ampakine CX614 reversed both respiratory depression and analgesia by fentanyl. Finally, Tübingen fish did not show respiratory depression by fentanyl. We therefore tested whether this strain is similarly insensitive to fentanyl in our analgesia assay. No difference in swimming velocity was observed between formalin and formalin/fentanyl group in TU fish (p=0.132), (*Figure 7C*), demonstrating that fentanyl did not reduce nociception elicited by formalin.

## Discussion

Opioid drugs are widely used as analgesics but present the severe side-effect of respiratory depression that can be lethal with overdose. No safe opioid therapy currently exists to treat severe and chronic pain. To identify new potent pain treatments without side-effects, opioid drug discovery needs to explore new research avenues. Novel approaches to drug discovery have been recently used including structure-based drug discovery using computer simulations (*Manglik et al., 2016*) or cell assays (*Winpenny et al., 2016*). Most drug discovery approaches use target-based approaches which exclude many drug candidates that do not specifically target proteins of interest. Here we propose phenotype-based drug discovery approaches in larval zebrafish to identify potent opioid pain therapies without the side-effects of respiratory depression. The larval zebrafish has emerged as a powerful model system for drug and gene discovery since it combines the biological complexity of in vivo models with a nervous system homolog to humans including the respiratory neural network. Such approach has the potential to validate many drug candidates without making any assumptions on their mechanisms of action or targets. Our drug discovery approach proposes to test whether drugs or combinations of drugs can be used as analgesics without the side-effects of respiratory depression and such strategy is of considerable interest (*Dahan et al., 2018*; *Montandon and Slutsky, 2019*). We combined several assays to test the respiratory depressant

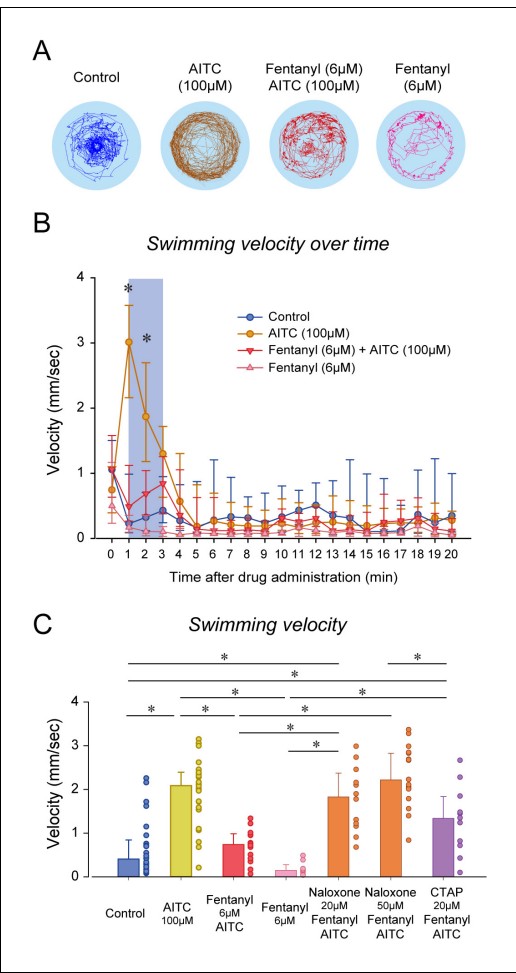

**Figure 6.** Nociception by AITC and analgesia by fentanyl in larval zebrafish. (**A**) As an index of nociception in larval zebrafish, we measured behavioral responses to the nociceptive stimulus AITC. We quantified swimming in freely-moving larvae. Swimming velocity was higher in zebrafish exposed to AITC (100 μM) compared to controls. (**B**) In fish exposed to fentanyl (6 μM) combined with AITC, swimming velocity was lower than in fish exposed to AITC alone. Larvae exposed to AITC showed an initial increase in velocity during the first 3 min of the recording, whereas larvae with AITC and fentanyl did not show a significant increase during the initial minutes. (**C**) Comparison of averaged velocities for the first 3 min (shaded blue in **B**b) after baseline (time = 0) showed that AITC (n = 26) significantly increased velocity compared to the control (n = 26), whereas fentanyl (n = 9) reduced the formalin response. Fentanyl alone did not significantly affect swimming velocity. Naloxone blocked the effect of fentanyl at 20 μM (n = 12) and 50 μM (n = 17). CTAP did not block the effect of fentanyl on nociception (n = 11). All data are presented as medians with error bars showing 25th and 75th percentile or interquartile range. Circles indicate individual data points for each zebrafish measured. * indicates groups significantly different with p<0.05. Source data can be found in *Figure 6—source data 1*.

The online version of this article includes the following source data and figure supplement(s) for figure 6:

**Source data 1.** Nociception by AITC and analgesia by fentanyl.
**Figure supplement 1.** Angular velocity in response to AITC and opioid drugs.

---

effects of pain killers and their analgesics properties. We demonstrated that larval zebrafish (days post-fertilization 12–14) can be used to test the analgesic properties of opioid drugs and to test the severity of respiratory depression. Our assays provide simple, yet effective, ways to quickly test potential drug candidates in a complex animal model while preserving its central nervous system and brain blood barrier. Due to the high production of fish embryos, we can use these phenotypic assays to perform high-throughput drug screening protocols that are otherwise not feasible in other animal models. Importantly, drug screening in rodents is not feasible as it would require extensive

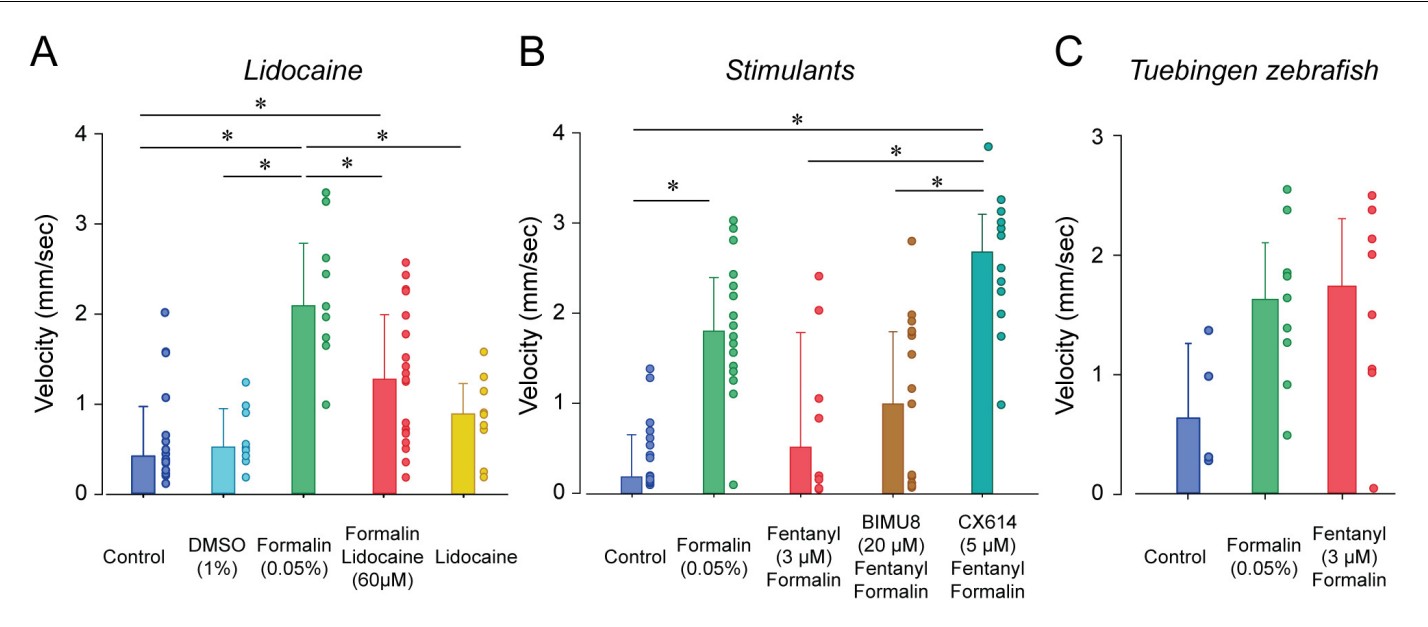

**Figure 7.** Analgesic profiles of larval zebrafish. (**A**) To determine whether the analgesia assay can be replicated with non-opioid analgesics, we used lidocaine, an analgesic widely used in the clinic. Lidocaine with formalin (n = 19) only moderately reduced swimming velocity compared to formalin alone (n = 10). (**B**) The respiratory stimulant BIMU8 administered with fentanyl and formalin (n = 17) did not significantly affect swimming velocity when compared to formalin/fentanyl (n = 8). On the other hand, CX614/formalin/fentanyl (n = 12) presented significant differences compared to the control (n = 21). (**C**) We also looked at the analgesic properties of fentanyl in Tübingen zebrafish. As observed with the respiratory assays, fentanyl (n = 10) did not reduce the swimming response compared to formalin alone (n = 9) as it did with AB zebrafish. All data are presented as medians with error bars showing 25th and 75th percentile or interquartile range. Circles indicate individual data points for each zebrafish measured. * indicates groups significantly different with p<0.05. Source data can be found in *Figure 7—source data 1*.

The online version of this article includes the following source data and figure supplement(s) for figure 7:

**Source data 1.** Analgesic profiles of larval zebrafish.

**Figure supplement 1.** Angular velocity in response to formalin, lidocaine, and stimulants.

resources and time. Here, we propose new behavioral assays in larval zebrafish where live animals with intact central nervous system can be leveraged for phenotype-based drug discovery combining respiratory depression and analgesia by opioid drugs.

Respiratory depression by opioids has been widely studied in mice, rats, dogs, and sheep (*Krause et al., 2009*; *Montandon et al., 2011*; *Montandon et al., 2016b*). The use of complex animal models to assess respiratory variables comes with many challenges. To accurately assess respiratory depression by opioid drugs, it is critical to quantify respiratory variables without the use of anesthetics which interact with the respiratory system, drug metabolism, and mechanisms of action of opioid drugs. Respiratory assessments in non-anesthetized rodents is the gold standard but it raises concerns due to the stress associated with animal handling, opioid injection, and changes due to altered arousal states and behaviours (*Montandon et al., 2016a*; *Montandon and Slutsky, 2019*), which may ultimately affect research outcomes. In larval zebrafish, we demonstrated, for the first time, that respiratory network activity can be assessed, that larval zebrafish present respiratory rate depression by opioid drugs in a similar fashion than humans, and that opioid pharmacology observed in humans is preserved in zebrafish. Most fishes have a complex respiratory system to move water through their gills. Although fishes use a different strategy to absorb oxygen and eliminate carbon dioxide than mammals, they rhythmically produce mandibular movements to move water through their gills. In lampreys, the paratrigeminal respiratory group (pTRG) generates rhythmic mandible movements (*Bongianni et al., 2016*). Because of their close evolutionary origins (*Missaghi et al., 2016*), the pTRG shares similarities with the mammalian respiratory network (*Cinelli et al., 2013*) which generates breathing and regulates respiratory depression by opioids (*Montandon et al., 2011*). The pTRG also presents similar functional properties than mammals (*Gray et al., 1999*) such as sensitivity to substance P (*Mutolo et al., 2010*) and to opioid drugs

(*Mutolo et al., 2007*). Here, we propose that respiratory mandible movements controlled, at least in part, by the pTRG can be used as an index of respiratory network activity, similarly to respiratory activity of the trigeminal muscle in mammals (*Jacquin et al., 1999*). We showed that the rate of respiratory movements presented dose-dependent decreases in response to increasing dosages of the MOR ligand fentanyl. Although it is difficult to compare administration of fentanyl through fish water in zebrafish and systemic injection in mammals, the dosages used in zebrafish corresponds to those used in rodents (*Yassen et al., 2008*). Respiratory depression by fentanyl was substantially blocked by the MOR antagonist naloxone, which is consistent with the fact that fentanyl is acting mostly through MORs (*James and Williams, 2020*). However, the highly selective MOR antagonist CTAP did not prevent respiratory depression by fentanyl, which is similar to the partial antagonism observed in some rodent studies (*Zhang et al., 2007*). These results suggest that either other opioid receptors are involved in the effect of fentanyl or that CTAP did not cross the blood brain barriers in zebrafish due to its large molecular weight. Interestingly, morphine reduced significantly respiratory rate in larval zebrafish at the low concentration of 1 µM, which is consistent with clinical studies (*Montandon et al., 2016a*), but not at higher concentrations. The lack of respiratory depression at high morphine concentrations may be due to disinhibition of excitatory networks regulating breathing in the zebrafish. In rodents, morphine disinhibits the ventral tegmental area (*Chen et al., 2015*), a brain area involved in sleep-wake states and arousal (*Venner et al., 2019*). Since increased arousal state reduces the severity of respiratory depression by opioids (*Montandon et al., 2016a*; *Montandon and Horner, 2019*), it may lead to a reduced respiratory depression at high concentrations of morphine.

The two major strains of wild-type zebrafish, AB and Tübinger (TU), showed different sensitivities to fentanyl. AB fish present pronounced respiratory depression, an effect not observed in TU fish, regardless of the concentration used (data not shown). This strain-specificity suggests that the mechanisms of action of opioid drugs differ between zebrafish strains which could be due to polymorphisms of the MOR gene as it can be found in humans (*Oertel et al., 2006*) or genetic differences in various genes involved in MOR inhibition (*Bian et al., 2012*). To determine whether respiratory stimulants can reverse respiratory depression by opioids in larval zebrafish, we tested two types of agonists targeting excitatory receptors: ampakines which comprise drugs acting on AMPA receptors (*Ren et al., 2006*) and serotoninergic agents (*Manzke et al., 2003*). CX-614, an ampakine allosteric modulator of AMPA receptors, reversed respiratory depression in our larval zebrafish models, which is consistent with its effects in rodents (*Ren et al., 2006*) or humans (*Dahan et al., 2018*). BIMU8, a 5-HT$_{4A}$ agonist, was administered in combination with fentanyl and compared with fentanyl alone. BIMU8 did not significantly reverse respiratory depression at the concentrations tested, which is consistent with the low efficacy of this treatment in humans (*Lötsch et al., 2005*). Other serotonin agents such as 5-HT$_1$ and 5-HT$_3$ agonists (*van der Schier et al., 2014*) can be easily tested using our zebrafish models.

Zebrafish larvae were previously used as animal models to study pain and analgesia. Acetic or citric acids were added to fish water to induce pain (*Lopez-Luna et al., 2017*; *Steenbergen and Bardine, 2014*). Our models did not use acids as nociceptive stimuli as they change water pH, to which zebrafish are sensitive (*Avdesh et al., 2012*; *Steenbergen and Bardine, 2014*). Instead, we used formalin, a nociceptive stimulus widely used to induce pain in rodent models (*Yoon et al., 2015*), and showed that formalin increased swimming velocity. Here, we propose that increased swimming velocity in response to formalin represents the fish escape response to nociceptive stimuli. In fact, lidocaine, a non-opioid analgesic, applied with formalin showed a significant reduction compared to formalin alone. Lidocaine induces analgesia by inactivating voltage-gated sodium channels in neurons, without activating opioid receptors (*Tetzlaff, 2000*), and these results support the concept that formalin induces pain in larval zebrafish. To promote analgesia in larval zebrafish, fentanyl was combined with formalin and it reduced the swimming response to formalin, therefore suggesting that it reduced pain. It could be suggested that the escape response observed in our assays is due to the effect of formalin on locomotor activity (not nociception) and that the reduced response with fentanyl may be due to inhibition of motor activity. However, fentanyl alone did not reduce swimming velocity compared to control larvae. To complement the nociceptive assay with formalin, we used AITC, also known as 'mustard oil', which acts on cation channel transient receptors potential ankyrin 1one and transient receptor potential vanilloid 1 (*Oda et al., 2016*), two receptors involved in nociception to chemical compounds (*Bamps et al., 2021*). Like formalin, the swimming response

to AITC was reduced by fentanyl. To determine the role of μ-opioid receptors in the analgesic responses to fentanyl, we administered the antagonist naloxone. In the formalin assay, naloxone did not block the effects of formalin, but blocked it in the AITC assay, which is consistent with the effects of naloxone in humans (*Gutstein, 2001*). Although naloxone did not significantly block analgesia when formalin was used, it is clear in *Figure 5d* that naloxone increased substantially velocity in some animals, but also reduced it in other animals. In summary, the fact that fentanyl reduced the nociceptive responses of AITC and formalin, two chemicals promoting nociception through different mechanisms, suggests that our approaches correctly assess opioid analgesia in larval zebrafish, while opioid pharmacology remains unclear in the formalin/fentanyl assays. As observed with the respiratory the CX-614 reversed analgesia which is consistent with their effects in rodents (*Dahan et al., 2018*). BIMU-8 did not reverse analgesia which is consistent with its lack of effects in clinical trials (*Dahan et al., 2018*). In conclusion, we established opioid analgesia assays that can be used to quantify pain and the analgesic properties of new opioid therapies that can easily be used for large scale drug discovery using larval zebrafish.

Opioid analgesics constitute essential pain therapies that present the lethal side-effect of respiratory depression therefore limiting their effective use in the clinical and at-home settings. There is currently no effective safe pain therapy due to the difficulty at identifying new drug combinations with potent analgesia but reduced respiratory side-effects (*Dahan et al., 2018*). Using larval zebrafish, we propose models allowing phenotype-based drug discovery approaches, i.e. permitting drug testing without assumptions related to mechanisms of action and targets. Our novel drug discovery models allow high-throughput drug screening in a simple and amenable animal model presenting similar pharmacological and genetic profiles than humans (*MacRae and Peterson, 2015*). Although zebrafish has been used to identify new anesthetics and their mechanisms of actions (*McGrath et al., 2020*; *Yang et al., 2019*), our study is the first to assess respiratory depression by opioid drugs in larval zebrafish combined with analgesia. In addition to high-throughput screening approaches, our models can also be used in combination of transgenic or knockout zebrafish to better understand the mechanisms opioid inhibition, analgesia and respiratory depression, as well as live microscopy of neural circuits of pain and respiration (*Ahrens et al., 2013*).

## Materials and methods

### Key resources table

| Reagent type (species) or resource | Designation | Source or reference | Identifiers | Additional information |
|---|---|---|---|---|
| Strain, strain background (*Danio rerio*) | AB Zebrafish | Hospital for Sick Children and the University of Toronto Mississauga (ON, Canada) | | |
| Strain, strain background (*Danio rerio*) | Tübingen (TU) Zebrafish | Hospital for Sick Children and the University of Toronto Mississauga (ON, Canada) | | |
| Strain, strain background | AB, TU cross zebrafish | Crossed at St. Michael's Hospital (ON, Canada) | | |
| Chemical compound, drug | Fentanyl citrate | Sandoz (QC, Canada) | Cat. No. 2520 | |
| Chemical compound, drug | Morphine sulfate | Sandoz (QC, Canada) | Cat. No. 5642 | |
| Chemical compound, drug | Naloxone hydrochloride | Omega (QC, Canada) | Cat. No. L0010224 | |
| Chemical compound, drug | CTAP (D-Phe-Cys-Tyr-D-Trp-Arg-Thr-Pen-Thr-NH$_2$) | Tocris (ON, Canada) | Cat. No. 1560/1 | |
| Chemical compound, drug | Lidocaine | Tocris (ON, Canada) | Cat. No. 3057 | |
| Chemical compound, drug | CX 614 | Tocris (ON, Canada) | Cat. No. 5149 | |
| Chemical compound, drug | BIMU 8 | Tocris (ON, Canada) | Cat. No. 4374/5 | |
| Chemical compound, drug | Allyl isothiocyanate (AITC) | Sigma-Aldrich (ON, Canada) | Cat. No. 36682 | |

*Continued on next page*

*Continued*

| Reagent type (species) or resource | Designation | Source or reference | Identifiers | Additional information |
|---|---|---|---|---|
| Chemical compound, drug | Formalin | VWR | Cat. No. 89370/094 | |
| Chemical compound, drug | Dimethyl sulfoxide (DMSO) | Sigma-Aldrich (ON, Canada) | Cat. No. 472301 | |
| Software, algorithm | Sigmaplot | Version 14, SAS | | |
| Software, algorithm | Adobe Illustrator | Creative Suite 5, Adobe | | |
| Software, algorithm | EthoVision XT | v15, Noldus Information Technology, Netherlands | | |
| Software, algorithm | Matlab | Mathworks, US | | |

## Animal husbandry

Animal practices and experiments in adult fish, larvae, and breeding pairs followed laboratory standards (*Avdesh et al., 2012*), and were carried out according to the procedures outlined by the Canadian Council on Animal Care and were approved by St. Michael's Hospital animal care committee. Only wildtype AB strain zebrafish larvae at 12–14 days post-fertilization (dpf) were used for experiments, except when comparisons between strains were made. Tübingen (TU) and crosses between TU and AB were used to compare strains. All experiments were performed in AB fish except when TU was specifically mentioned. All fish were housed on a 14/10 hr light/dark cycle and kept at a constant water temperature of 28°C ± 0.5°. Larvae and adults were originally obtained from the Hospital for Sick Children and the University of Toronto Mississauga (Toronto, ON). For breeding, male and female adult fish at 4 months of age were placed in a breeding tank and separated by a divider. The next day the divider was removed, and fish mated within the first half hour of the lights-on period and were returned to the rack (Aquaneering, CA, United States). The eggs were collected and placed in petri-dish filled with E2 embryo medium (NaCl, 15.0 mM; KCl, 0.5 mM; MgSO$_4$, 1.0 mM; CaCl$_2$, 1.0 mM; Na$_2$HPO$_4$, 0.05 mM; KH$_2$PO$_4$, 0.15 mM; NaHCO$_3$, 0.7 mM). Unfertilized eggs were removed.

Beginning at 5 days post-fertilization, larvae were fed with Ziegler AP100 (artificial plankton) dry larval diet (100 microns) and were moved to 0.8 litre tanks filled with system water, with a density of 20 fish/100 mL and were kept there until the experiment. Water quality was kept at a pH of 7.5 ± 0.5 and with a conductivity of 500-1000ppm. Dissolved oxygen was maintained at 6-7ppm. Nitrites and ammonia were kept at <150 ppm. All experiments were undertaken during daylight hours and fish were placed in an incubator at 28.5 ± 0.5°C.

## Drug treatments

All opioid drugs were used with Health Canada approval. Fentanyl citrate and morphine sulfate were obtained from Sandoz (QC, Canada). Naloxone hydrochloride was obtained from Omega (QC, Canada). The MOR antagonist CTAP (D-Phe-Cys-Tyr-D-Trp-Arg-Thr-Pen-Thr-NH$_2$), lidocaine, the Ampakine CX614, and the 5-HT$_{4A}$ receptor agonist BIMU8 were obtained from Tocris (ON, Canada). AITC was obtained from Sigma-Aldrich (ON, Canada).

## Determination of mandibular respiratory movements

Respiratory mandible movements were quantified in live zebrafish larvae using a custom-made system including a 4K high-definition camera (Basler 4K, 4096pi x 3000pi, model acA4112-30um, Edmunds Optics) and partially telecentric 7x zoom lens (Edmund Optics). The water was kept at a constant 28.5 ± 0.5°C temperature using a heating pad placed under the multi-well plate. With this system, simultaneous recordings of 12 zebrafish larvae can be performed. As an index of respiratory network activity, mandible movements were measured, and respiratory rate was quantified (*Figure 1A*). In a custom-made 12-well clear plate, larval zebrafish were placed in wells (diameter 10 mm, depth 4 mm) containing 54 µL of embryo medium. Using our video recording system, a dorsal video was performed for each drug combination. We used two approaches to determine the rate of mandible movement. Rate of mandible movements was visually counted by two independent researchers, blind to the drug combinations. Using this approach, the number of mandible

movements per minute was calculated for each condition and each animal. In a subset of zebrafish larvae, mandible movements were also quantified by measuring pixel intensity changes in a defined area around the fish head. Tracings of pixel intensity over time were plotted and the rate of intensity changes was quantified using a custom-made software in Matlab (Mathworks, US). The validity of visual quantification was then compared to software quantification.

## Respiratory depression by opioids in zebrafish

Zebrafish larvae were positioned in wells and were left for 10 min to acclimatize to the environment. A video of baseline respiratory movement was then taken for 1 min. Following the baseline recordings, combinations of drugs were applied to each well in a volume of 6 µL per well. VA video was recorded for 30 min following drug administration. To determine whether various drugs induce respiratory rate depression, we compared the rates of mandible movements between several groups of larvae: control group (embryo medium), fentanyl group, fentanyl/naloxone group (combination of fentanyl and naloxone), fentanyl/other drugs (fentanyl and potential drug candidates). When possible, animals from different groups were tested simultaneously in 12-well plates (for example: four larvae per group with three groups were tested). However, several separate recordings of different 12-well plates were necessary to provide the correct number of animals per group. Respiratory rate values collected in larval zebrafish were normalized according to baseline mandible rates acquired before drugs were administered. Normalization considerably reduced the high variability encountered in larval zebrafish. Larval zebrafish may present different growth and behaviors at 12-14dpf due to variable development rates and access to food and nutrients within the tank. To consistently select larvae at similar development stages, we established selection criteria. For respiratory assays, larvae with rates of mandible movements below 60 and above 160 movements/min at baseline (before drugs were administered) were not considered. Such exclusion criteria considerably reduce the inter-group variability observed in zebrafish larvae.

## Swimming behaviour quantification

Nociception was assessed by measuring the swimming escape response to nociceptive stimuli. Zebrafish were placed individually into wells of a custom-made 24-well plate containing 90 µL of embryo water. The well plate was made of white acrylic with a back-light to allow video recording. The plate was heated at 28.5 ± 0.5°C. The same custom-made video recording apparatus, as mentioned above, was used to record swimming in larval zebrafish. Swimming experiments were analyzed using a commercial software (EthoVision XT v15, Noldus Information Technology, Netherlands), and swimming velocity (mm/sec) and angular velocity (degrees/sec) were quantified. Before applying drugs to the plate wells containing larvae, we measured a 10 min baseline. Only larvae with velocity between 0.1 and 2 mm/sec during baseline (before drugs were administered) were considered for analysis. Formalin or (allyl)-isothiocyanate (AITC) were then used as nociceptive stimuli. Such stimuli were previously validated in mammalian and adult zebrafish (*Magalhães et al., 2017*; *Oda et al., 2016*) studies. AITC directly acts on transient receptor potential A1 which is involved in pain response. We applied formalin or AITC in combination with opioid analgesics to determine opioid analgesia in larval zebrafish.

## Opioid analgesia in larval zebrafish

Zebrafish larvae (12–14 days post-fertilization) were positioned in wells of a 24-well plate and left 10 min to acclimatize. A video of the baseline movements was taken for 10 min. Following baseline recordings, combinations of drugs were applied to each well in a volume of 10 µL and video was recorded for 15 min. To determine whether various drug combinations induce analgesia, we compared the swimming velocity between several groups of larvae: control group (embryo medium was administered), formalin or AITC group, fentanyl/formalin or fentanyl/AITC group (combination of fentanyl and formalin), formalin/fentanyl/naloxone or AITC/fentanyl/naloxone. CTAP was also used instead of naloxone. When possible, these groups were tested simultaneously in 24-well plates (six larvae per group if four groups were tested). Multiple recordings were needed to provide the correct number of animals per group. The swimming and angular velocities were calculated for the first 3 min following treatment (minutes 0–3) and were used for analysis.

## Statistical analysis

Data in larval zebrafish do not always follow a normal distribution. Therefore, normality was tested using a Shapiro-Wilk method and equality of variances using the Brown-Forsythe test. All respiratory data presented in the Results section followed normal distribution when respiratory rate was normalized according to baseline respiratory rate (before any drugs were administered). When data followed a normal distribution, we compared multiple groups using a one-way ANOVA followed by Holm-Sidak post-hoc tests to compare individual groups. We presented individual data points, group means and standard deviation as error bars. However, data in larval zebrafish are often not normally distributed. For instance, swimming velocity and angular velocity did not follow normal distribution, even when normalized according to baseline. When data distribution was not normal, we compared experimental groups using Wilcoxon rank tests and All Pairwise Multiple Comparison Procedures (Dunn's method) as post-hoc tests. We presented data as individual data points and medians with error bars showing 25th and 75th percentiles (interquartile range). In some cases, data presented clear outliers. To objectively identify outliers, we selected all data points below and above 1.5 x the interquartile range and eliminated these data points from the data set (*Michel et al., 2020*). All statistical tests and graphs were done using Sigmaplot (version 14, SAS) and figures were prepared with Adobe Illustrator (Creative Suite 5, Adobe).

## Acknowledgements

Research was supported by the Research Innovation Council Fund of St. Michael's Hospital Foundation and the JP Bickell Foundation Medical Grant. SZ was supported by a CIHR Canada Graduate Scholarship. We thank Alexandra Ho for her help with data analysis.

## Additional information

### Funding

| Funder | Grant reference number | Author |
|---|---|---|
| St. Michael's Hospital Foundation | RIC | Gaspard Montandon |
| J. P. Bickel Foundation | Medical Grant | Gaspard Montandon |

The funders had no role in study design, data collection and interpretation, or the decision to submit the work for publication.

### Author contributions

Shenhab Zaig, Conceptualization, Data curation, Formal analysis, Validation, Investigation, Visualization, Methodology, Writing - original draft; Carolina da Silveira Scarpellini, Formal analysis, Validation, Investigation, Methodology; Gaspard Montandon, Conceptualization, Formal analysis, Supervision, Funding acquisition, Validation, Investigation, Visualization, Methodology, Writing - original draft, Project administration

### Author ORCIDs

Carolina da Silveira Scarpellini ![ORCID] http://orcid.org/0000-0001-5576-3468
Gaspard Montandon ![ORCID] https://orcid.org/0000-0003-3587-4472

### Ethics

Animal experimentation: The protocol was approved by the Animal Care Committee of St. Michael's Hospital. Protocol: ACC-811.

### Decision letter and Author response

Decision letter https://doi.org/10.7554/eLife.63407.sa1
Author response https://doi.org/10.7554/eLife.63407.sa2

## Additional files

### Supplementary files
- Transparent reporting form

### Data availability
All data generated during this study are included in the manuscript. Source files are available.

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
