## [Decision Letter]

**Acceptance summary:**

Although opioids, e.g., morphine, remain the major pharmacotherapy for severe, chronic pain, adverse side effects, especially respiratory depression, significantly limit their utility. Here, Zaig et al. established a zebrafish model for mammalian opioid-induced respiratory depression. This model will enable future work to efficiently and effectively screen for novel compounds that provide opioid receptor-mediated analgesia but not respiratory depression.

**Decision letter after peer review:**

Thank you for submitting your article "Respiratory depression and analgesia by opioid drugs in freely-behaving larval zebrafish" for consideration by *eLife*. Your article has been reviewed by three peer reviewers, and the evaluation has been overseen by a Reviewing Editor and Michael Taffe as the Senior Editor. The following individual involved in review of your submission has agreed to reveal their identity: Jan Marino Ramirez (Reviewer #3).

The reviewers have discussed the reviews with one another and the Reviewing Editor has drafted this decision to help you prepare a revised submission.

Essential revisions:

In this report Zaig describe a zebrafish model for mammalian opioid induced respiratory depression, as a potential screening approach for novel, selective opioids. All reviewers found the manuscript of significant interest, however, there are several concerns that need to be addressed and a limited requirement for some additional experimental information. Certainly the conclusion needs to be modified. Specifically, the reviewers believe that you have overstated the similarities between mammals and zebrafish. The mandibular rhythm in zebrafish is most certainly not generated by the preBötC, but more likely is by an oscillator in the pons. Thus, the fact that both movements are opioid sensitive doesn't mean that they are sensitive in the same way.

A major concern is that the same data are recycled throughout the paper. This is not how the experiments are described in the Materials and methods and this is also not how the supplementary data is annotated. Different animals are annotated to have the same frequency of mandibular movements. If there are other controls that were actually used, then the correct comparisons are not being performed. This is something the authors can address or correct without new data.

An experiment that should be reported in the course of revision relates to the results described in Figure 5: Here, CTAP does not show a rescue (perhaps a trend), while naloxone does. Therefore, a naloxone only control is required to conclude that the fentanyl rescue is not due to antagonizing endogenous opioid signaling on δ or κ receptors. Additionally, a fentanyl only control in Figure 5B and C would be important to show.

Also, please list the concentration of naloxone used in these experiments. Related to this point: All of the fentanyl data for Figure 5 have been collected, but not plotted in Figure 5B and C.

Additional comments to be addressed:

There are discrepancies between the methods description and source data provided.

Specifically, the Materials and methods state that "To determine whether various drugs induce respiratory rate depression, we compared the rates of mandible movements between several groups of larvae: control group (embryo medium), fentanyl group, fentanyl/naloxone group (combination of fentanyl and naloxone), fentanyl/other drugs (fentanyl and potential drug candidates). All these groups were tested simultaneously in 12-well plates (4 larvae per group if 3 groups were tested). By combining all groups in the same multi-well plate, we ensured that all animals experienced the same conditions therefore allowing comparisons between groups." In fat, the number of control fentanyl experiments used in each experiment is different from the fentanyl+drug numbers (for example, Figure 4B). How can this be if these are done all as replicates with similar number per 12 well plate?

Additionally, the source data show that the fentanyl control data are exactly the same between experiments, Figures 3E, 4A, B, and probably 3B (based on the figure). Also, these data are mostly the same as Figure 2E. However, in the source data, each of these exact same data points has a different fish number annotated. Are these the same fish or different fish? There is an error somewhere and this is absolutely critical to explain since the comparison cohort impacts every conclusion made about mandibular movement. Conclusions shown to be statistically significant or "tending that way" may no longer be valid conclusions.

A central hypothesis is that mandibular movement produced by the paratrigeminal group represents the respiratory rhythm created by the preBötC. However, two successful pharmacological approaches that rescue opioid depression of the preBötC in rodents, 5HT4a (BIMU8, Mazke et al.) and ampakine therapy (CX614, Lorier et al.), do not significantly rescue fentanyl suppression of mandibular jaw movement (Figure 4). Please address these findings.

Although the data may trend towards significance, there is a concern about the comparison cohort used. The 1µM fentanyl data used in Figure 3B, E and Figure 4A, B appears to be the same, but different than or partially overlapping with the fentanyl cohort in Figure 2E. Since the "% baseline change" in Figure 2E is less, if these data are included in Figures 3 and 4, these trending rescues would become less obvious. A clearer experiment might be to attempt rescue of mandibular movement in the same animal after measuring its mandibular depression.

---

## [Author Response]

Essential revisions:In this report Zaig describe a zebrafish model for mammalian opioid induced respiratory depression, as a potential screening approach for novel, selective opioids. All reviewers found the manuscript of significant interest, however, there are several concerns that need to be addressed and a limited requirement for some additional experimental information. Certainly the conclusion needs to be modified. Specifically, the reviewers believe that you have overstated the similarities between mammals and zebrafish. The mandibular rhythm in zebrafish is most certainly not generated by the preBötC, but more likely is by an oscillator in the pons. Thus, the fact that both movements are opioid sensitive doesn't mean that they are sensitive in the same way.

We agree with the reviewers. Our study only shows that the respiratory network involved in breathing in fish is depressed by opioid drugs in a similar way than in mammals. The precise circuits inhibited by opioids are still unknown in zebrafish. We have changed the text in the Introduction related to pTRG and preBötC.

A major concern is that the same data are recycled throughout the paper. This is not how the experiments are described in the Materials and methods and this is also not how the supplementary data is annotated. Different animals are annotated to have the same frequency of mandibular movements. If there are other controls that were actually used, then the correct comparisons are not being performed. This is something the authors can address or correct without new data.

The reviewers are right. Some data have been used in different figures when the experiments should have been separately performed. To avoid any confusions, we have decided to repeat all these experiments highlighted by the reviewers. We repeated experiments as described above.

An experiment that should be reported in the course of revision relates to the results described in Figure 5: Here, CTAP does not show a rescue (perhaps a trend), while naloxone does. Therefore, a naloxone only control is required to conclude that the fentanyl rescue is not due to antagonizing endogenous opioid signaling on δ or κ receptors.

We agree. One of the challenges using formalin is that it substantially increases variability in swimming. Although it is visible that naloxone/formalin/fentanyl substantially increase swimming, it also increases variability of the data which makes the average velocity not significant different. However, when using AITC, the variability doesn’t increase as much and naloxone was able to reverse the analgesia induced by fentanyl. It is now addressed in the Discussion. We also performed additional experiments with naloxone and CTAP alone at various concentrations.

Additionally, a fentanyl only control in Figure 5B and C would be important to show.

We have now included fentanyl in Figure 5 showing time course and representative tracings.

Also, please list the concentration of naloxone used in these experiments.

It has been included.

Related to this point: All of the fentanyl data for Figure 5 have been collected, but not plotted in Figure 5B and C.

We performed new experiments and now provide new data.

Additional comments to be addressed:There are discrepancies between the methods description and source data provided.Specifically, the Materials and methods state that "To determine whether various drugs induce respiratory rate depression, we compared the rates of mandible movements between several groups of larvae: control group (embryo medium), fentanyl group, fentanyl/naloxone group (combination of fentanyl and naloxone), fentanyl/other drugs (fentanyl and potential drug candidates). All these groups were tested simultaneously in 12-well plates (4 larvae per group if 3 groups were tested). By combining all groups in the same multi-well plate, we ensured that all animals experienced the same conditions therefore allowing comparisons between groups." In fat, the number of control fentanyl experiments used in each experiment is different from the fentanyl+drug numbers (for example, Figure 4B). How can this be if these are done all as replicates with similar number per 12 well plate?

We have modified the Materials and methods to make it clearer. We tried to have all the conditions tested at the same time in the 12-well plate. However, there were more groups and animals per group than the number of wells per plate (12). We therefore had to combine recordings from various 12-well plates. Also, due to the criteria we used to exclude fish, the number of fish measured for each condition varies considerably. After revisions of the data and new experiments performed, no recordings have been used more than once. All experimental data are now unique for each panel in the figures.

Additionally, the source data show that the fentanyl control data are exactly the same between experiments, Figures 3E, 4A, B, and probably 3B (based on the figure). Also, these data are mostly the same as Figure 2E. However, in the source data, each of these exact same data points has a different fish number annotated. Are these the same fish or different fish? There is an error somewhere and this is absolutely critical to explain since the comparison cohort impacts every conclusion made about mandibular movement. Conclusions shown to be statistically significant or "tending that way" may no longer be valid conclusions.

We repeated the experiments, carefully checked all the data and updated the spreadsheet accordingly. Data points of each condition are unique and have never been used more than once.

A central hypothesis is that mandibular movement produced by the paratrigeminal group represents the respiratory rhythm created by the preBötC. However, two successful pharmacological approaches that rescue opioid depression of the preBötC in rodents, 5HT4a (BIMU8, Mazke et al.) and ampakine therapy (CX614, Lorier et al.), do not significantly rescue fentanyl suppression of mandibular jaw movement (Figure 4). Please address these findings.

We agree with the reviewers. Although BIMU-8 was successful in some animal studies it had limited success in humans. In our experiments, BIMU-8 did not reverse respiratory depression, which may be due to differences in the serotoninergic system of zebrafish or differences in drug diffusion through the fish brain blood barrier. We address it in the Discussion.

Although the data may trend towards significance, there is a concern about the comparison cohort used. The 1µM fentanyl data used in Figure 3B, E and Figure 4A, B appears to be the same, but different than or partially overlapping with the fentanyl cohort in Figure 2E. Since the "% baseline change" in Figure 2E is less, if these data are included in Figures 3 and 4, these trending rescues would become less obvious. A clearer experiment might be to attempt rescue of mandibular movement in the same animal after measuring its mandibular depression.

We have performed additional experiments to answer these questions. We do not perform sequential experiments with several conditions in the same fish, because drug pipetting and movement of the multi-well plate affect mandible and behavioral recordings.